# INVARIANT AGGREGATOR FOR DEFENDING AGAINST FEDERATED BACKDOOR ATTACKS

## ABSTRACT

Federated learning is gaining popularity as it enables training of high-utility models across several clients without directly sharing their private data. As a downside, the federated setting makes the model vulnerable to various adversarial attacks in the presence of malicious clients. Specifically, an adversary can perform backdoor attacks to control model predictions via poisoning the training dataset with a trigger. In this work, we propose a mitigation for backdoor attacks in a federated learning setup. Our solution forces the model optimization trajectory to focus on the invariant directions that are generally useful for utility and avoid selecting directions that favor few and possibly malicious clients. Concretely, we consider the sign consistency of the pseudo-gradient (the client update) as an estimation of the invariance. Following this, our approach performs dimension-wise filtering to remove pseudo-gradient elements with low sign consistency. Then, a robust mean estimator eliminates outliers among the remaining dimensions. Our theoretical analysis further shows the necessity of the defense combination and illustrates how our proposed solution defends the federated learning model. Empirical results on three datasets with different modalities and varying number of clients show that our approach mitigates backdoor attacks with a negligible cost on the model utility.

## 1 INTRODUCTION

Federated learning enables multiple distrusting clients to jointly train a machine learning model without sharing their private data directly. However, a rising concern in this setting is the ability of potentially malicious clients to perpetrate backdoor attacks. To this end, it has been argued that conducting backdoor attacks in a federated learning setup is practical (Shejwalkar et al., 2022) and can be effective (Wang et al., 2020). For instance, the adversary can connect to a federated learning system as a legitimate user and conduct a backdoor attack that forces the model to mispredict. The impact of such attacks is quite severe in many mission-critical federated learning applications. For example, anomaly detection is a common federated learning task where multiple parties (e.g., banks or email users) collaboratively train a model that detects frauds or phishing emails. Backdoor attacks allow the adversary to successfully circumvent these detection methods.

The most common backdoor attack embeds *triggers* in the data samples and forces the model to make an adversary-specified prediction when the trigger is observed (Liu et al., 2018; Bagdasaryan et al., 2020). Thus, an adversary can conduct a backdoor attack by generating a trigger that statistically correlates with a particular label. Once the adversary injects these trigger-embedded backdoor data samples into the training data, the model can entangle the trigger-label correlation and predict as the adversary specifies. Meanwhile, the backdoor attack often does not degrade the predictive accuracy on the benign samples, making backdoor detection difficult in practice (Wang et al., 2020).

In federated learning, the server aggregates only the client-level updates (a.k.a. pseudo-gradient or gradient for short) without control over the training procedure or any data samples. Such limited visibility of the federated learning server on the client-side training makes defending against backdoor attacks challenging. Common defenses against backdoor attacks aim at identifying the backdoor data samples or poisoned model parameters and usually require access to at least a subset of the training data (Tran et al., 2018; Li et al., 2021a), which is prohibitive for a federated learning server. Other defense methods against untargeted poisoning attacks that degrade the model utility

(Shejwalkar et al., 2022) are applicable but lack robustness against backdoor attacks, as discussed in Section 6.2.

**Our approach.** Our defense leverages the observation that learning from the poisonous data does not benefit the model on benign data and vice versa. Therefore, focusing on the invariant directions that are generally beneficial in the model optimization trajectory helps defending against the aforementioned backdoor attack (which often lead to non-invariant directions). To this end, we develop a defense by examining each *dimension* of the gradients on the server-side and checking whether the dimension-wise gradients point in the same *direction* across the clients. Here, a dimension-wise gradient can point to a positive or negative direction, or have a zero value. In the case of small learning rates and for a specific dimension, two gradients pointing in the same direction means that taking the direction of one gradient can benefit the other. As such, the *invariance* of a direction depends on how many dimension-wise gradients align with that direction. Following this intuition, we define the *sign consistency* of a dimension by the average gradient sign. The higher the sign consistency is, the more invariant direction the gradient dimension may have.

Designing such a method carefully selecting only the invariant gradient directions is non-trivial, especially given the non-i.i.d. gradient distributions across benign clients and the presence of malicious clients. Hence, our approach enforces two separate treatments for each gradient dimension. First, we employ an AND-mask (Parascandolo et al., 2021), a dimension-wise filter setting the gradient dimension with sign consistency below a given threshold to zero. However, this alone is not enough: the malicious clients can still use outliers to mislead the aggregation result in the remaining highly consistent dimensions. To address this issue, we propose using the trimmed-mean estimator (Xie et al., 2020b; Lugosi & Mendelson, 2021), as a means to remove the outliers. Our analysis suggests that the AND-mask complements the trimmed-mean estimator well, motivating their composition.

We support the proposed approach with a theoretical analysis under a conventional linear regime (Rosenfeld et al., 2021; Wang et al., 2022; Zhou et al., 2022; Manoj & Blum, 2021), showing that the composition of the AND-mask and the trimmed-mean estimator is necessary for defending against backdoor attacks. Our analysis starts with feature invariance and discusses the connection between feature invariance and gradient sign consistency. Then, we outline conditions under which trigger-based backdoor attacks can lead to non-invariant directions and decrease the sign consistency of a dimension. Further analysis results demonstrate the necessity for the combination of both the AND-mask and the trimmed-mean estimator. Simulation results in Appendix D.1 further verify our theoretical results.

Our empirical evaluation employs the strong edge-case backdoor attack (Wang et al., 2020), as detailed in Section 6.1, to test our defense. Empirical results on tabular (phishing emails), visual (CIFAR-10) (Krizhevsky, 2009; McMahan et al., 2017), and text (Twitter) (Caldas et al., 2018) datasets demonstrate that our method is effective in defending against backdoor attacks without degrading utility as compared to prior works. On average, our approach decreases the model accuracy on backdoor samples by 61.6% and only loses 1.2% accuracy on benign samples compared to the standard FedAvg aggregator (McMahan et al., 2017).

**Contributions.** Our contributions are as follows:

- We develop a combination of defenses using an AND-mask and the trimmed-mean estimator against the backdoor attack by focusing on the dimension-wise invariant directions in the model optimization trajectory.

- We theoretically analyze our strategy and demonstrate that a combination of an AND-mask and the trimmed-mean estimator is necessary in some conditions.

- We empirically evaluate our method on three datasets with varying modality, model architecture, and client numbers, as well as comparing the performance to existing defenses.

## 2 RELATED WORK

**Backdoor Attack.** Common backdoor attacks aim at misleading the model predictions using a trigger (Liu et al., 2018). The trigger can be digital (Bagdasaryan et al., 2020), physical (Wenger et al., 2021), semantic (Wang et al., 2020), or invisible (Li et al., 2021b). Recent works extended

backdoor attacks to the federated learning setting and proposed effective improvements such as gradient scaling (Bagdasaryan et al., 2020) or generating edge-case backdoor samples (Wang et al., 2020). The state-of-the-art edge-case backdoor attack shows that using backdoor samples with low probability density on benign clients (i.e., unlikely samples w.r.t. the training distribution) are hard to defend in the federated learning setting.

**Centralized Defense.** There is a line of work proposing centralized defenses against backdoor attacks where the main aim is either detecting the backdoor samples (Tran et al., 2018) or purifying the model parameters that are poisoned (Li et al., 2021a). However, applying such centralized defense to federated learning systems is in practice infeasible due to limited access to the client data in many implementations.

**Federated Defenses.** Several recent works have attempted to defend against backdoor attacks in federated learning systems. Sun et al. (2019) shows that weak differential-private (weak-dp) federated averaging can mitigate the backdoor attack. However, the weak-dp defense is circumvented by the improved edge-case federated backdoor attack (Wang et al., 2020). Nguyen et al. (2021) suggest that the vector-wise cosine similarity can help detect malicious clients performing backdoor attacks. The vector-wise cosine similarity is insufficient when the backdoor attacks can succeed with few poisoned parameters, incurring little vector-wise difference (Wu & Wang, 2021). Other defenses against untargeted poisoning attacks (Blanchard et al., 2017; Xie et al., 2020b) lack robustness against the backdoor attack. Sign-SGD with majority vote (Bernstein et al., 2018; 2019) is similar to our approach, but it always takes the majority direction instead of focusing on the invariant directions. Section 6.2 discusses the limitation of previous defenses in more detail, along with the empirical evaluation. Unlike existing works, our defense encourages the model to pursue invariant directions in the optimization procedure.

**Certification.** Unlike the above discussed defenses, certification (Xie et al., 2021) aims at extinguishing backdoor samples within a neighborhood of a benign sample. A direct comparison between certification and our defense is not meaningful due to the different evaluation metrics. Certification considers the certification rate of benign samples as the metric, while our defense aims at reducing the accuracy of the backdoor samples. However, it would be interesting to investigate whether the proposed defense can ease the certification of a model.

## 3 PROBLEM SETUP

**Notation.** We assume a synchronous federated learning system, where $N$ clients collaboratively train an ML model $f : \mathcal{X} \to \mathcal{Y}$ with parameter $\boldsymbol{w}$ coordinated by a server. An input to the model are the data samples $\boldsymbol{x} \in \mathcal{X} = \mathbb{R}^{\mathrm{d}}$ with $d$ features indexed by $k$ and a label $y$. There are $N' < \frac{N}{2}$ adversarial clients aiming at corrupting the ML model during training (Shejwalkar et al., 2022). The $i^{\mathrm{th}}$, $i \in [1, ..., N]$, client has $n_i$ data samples, being benign for $i \in [1, ..., N - N']$ or being adversarial for $i \in [N - N' + 1, ..., N]$. The synchronous federated learning is conducted in $T$ rounds. In each round $t \in [1, ..., T]$, the server broadcasts a model parameterized by $\boldsymbol{w}_{t-1}$ to all the participating clients. We omit the subscript $t$ while focusing on a single round. Then, the $i^{\mathrm{th}}$ client optimizes $\boldsymbol{w}_{t-1}$ on their local data samples indexed by $j$ and report the locally optimized $\boldsymbol{w}_{t,i}$ to the server. We define pseudo-gradient $\boldsymbol{g}_{t,i} = \boldsymbol{w}_{t-1} - \boldsymbol{w}_{t,i}$ being the difference between the locally optimized model and the broadcasted model from the previous round. Note, for simplicity, that we often use the term "gradient" to refer to the pseudo-gradient. Once all gradients are uploaded, the server aggregates them and produces a new model with parameters $\boldsymbol{w}_t$ using the following rule: $\boldsymbol{w}_t = \boldsymbol{w}_{t-1} - \sum_{i=1}^{N} \frac{n_i}{\sum_{i=1}^{N} n_i} \boldsymbol{g}_{t,i}$. The goal of federated learning is to minimize a weighted risk function over the $N$ clients: $F(\boldsymbol{w}) = \sum_{i=1}^{N} \frac{n_i}{\sum_{i=1}^{N} n_i} F_i(\boldsymbol{w}) = \sum_{i=1}^{N} \frac{n_i}{\sum_{i=1}^{N} n_i} \mathbb{E}_{\mathcal{D}_i}[\ell(f(x; \boldsymbol{w}), y)]$, where $\ell : \mathbb{R} \times \mathcal{Y} \to \mathbb{R}$ is a loss function. $\odot$ denotes the Hadamard product operator.

**Threat Model.** The adversary generates a backdoor data sample $\boldsymbol{x}'$ by embedding a trigger in a benign data sample $\boldsymbol{x}$ and correlating the trigger with a label $y'$, which is different from the label $y$ of the benign data sample. We use $\mathcal{D}'$ to denote the distribution of backdoor data samples. Then, the malicious clients connect to the federated learning system and insert backdoor data samples into the training set. Since the goal of federated learning is to minimize the risk over all clients' datasets, the model can entangle the backdoor while trying to minimize the risk over backdoor samples on the malicious clients. Appendix A visualizes some backdoor samples.

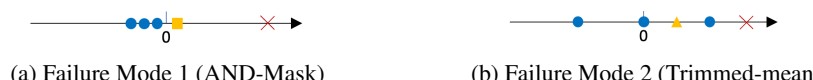

(a) Failure Mode 1 (AND-Mask)  (b) Failure Mode 2 (Trimmed-mean)

Figure 1: Failure modes of AND-mask (a) and trimmed-mean estimator (b). Note that an aggregator fails if the malicious value flips the sign of the aggregation result compared to the true aggregate. Blue dots are benign values. Red crosses are malicious values. The orange box is an arithmetic mean and the orange triangle is a trimmed-mean. In (a), the adversary uses outliers to flip the sign of the arithmetic mean when the benign values have the same sign. (b) shows that the trimmed-mean estimator may bias toward the malicious value when the average is supposed to be zero, but the benign values have diverse signs.

To simplify tedious notation, we assume all users have the same number of samples i.e., $n_i = n_{i'}, \forall i \neq i'$.

## 4 METHOD

This section presents the proposed server-side defense operating on the gradients. We start with an overview of the idea, then introduce the two complementary components of our defense and outline the invariant aggregator steps.

### 4.1 OVERVIEW

In the proposed defense, we aim at finding invariant directions to optimize the federated learning model, such that the model is generally utilitarian for most of clients and can exclude the directions that benefit a subset of potentially malicious clients. Since our defense operates with the gradients, we consider the invariant direction from a first-order perspective (Parascandolo et al., 2021). Expanding the loss function around the current weight $\boldsymbol{w}$ on client $i$, for a parameter update $\boldsymbol{g}$, we have:

$$\mathbb{E}_{\mathcal{D}_i}[\ell(\boldsymbol{x}, y; \boldsymbol{w} + \boldsymbol{g})] = \mathbb{E}_{\mathcal{D}_i}[\ell(\boldsymbol{x}, y; \boldsymbol{w})] + \mathbb{E}_{\mathcal{D}_i}[\nabla_{\boldsymbol{w}} \ell(\boldsymbol{x}, y; \boldsymbol{w})]^\top \boldsymbol{g} + R_2(\boldsymbol{w} + \boldsymbol{g}), \quad (1)$$

where $R_2(\boldsymbol{w} + \boldsymbol{g})$ is a second-order Taylor remainder. With a reasonably small $\|\boldsymbol{g}\|$ (achievable with small learning rates), the remainder term $R_2(\boldsymbol{w} + \boldsymbol{g})$ is negligible and the change of the loss function mainly depends on the first-order gradient and the parameter update $\boldsymbol{g}$. Since learning the trigger benefits the malicious clients exclusively, there exists at least one gradient dimension $k$ and one benign client $i \in \{1, ..., N - N'\}$ where $\mathbb{E}_{\mathcal{D}'}[\nabla_{\boldsymbol{w}_k} \ell(\boldsymbol{x}, y; \boldsymbol{w})] \neq 0$ and the dimension-wise gradient have inconsistent signs, i.e.,

$$\mathbb{E}_{\mathcal{D}_i}[\nabla_{\boldsymbol{w}_k} \ell(\boldsymbol{x}, y; \boldsymbol{w})] \times \mathbb{E}_{\mathcal{D}'}[\nabla_{\boldsymbol{w}_k} \ell(\boldsymbol{x}', y'; \boldsymbol{w})] \leq 0. \quad (2)$$

If the condition in Equation 2 is not true, then learning the trigger would always benefit the model on benign data, thereby contradicting common empirical observation. A more detailed analysis of the condition in Equation 2 is in Section 5.2. The proposed methods show how to treat the inconsistent signs and defend against the backdoor attack by only allowing invariant directions. A dimension-wise analysis is necessary because backdoor attacks can succeed by poisoning few inconsistent dimensions (Wu & Wang, 2021) without incurring much vector-wise difference, as Section 6.2 will show.

We consider two treatments for each gradient dimension to help the model avoid the direction specified by $\mathbb{E}_{\mathcal{D}'}[\nabla_{\boldsymbol{w}} \ell(\boldsymbol{x}', y'; \boldsymbol{w})]$: (Treatment 1) setting the dimension with inconsistent signs to zero using an AND-mask (Parascandolo et al., 2021) such that no client benefits or (Treatment 2) employing a robust mean estimator (e.g., trimmed-mean (Xie et al., 2020b)) to remove the malicious values that cause the inconsistent sign. To achieve a better result, we combine these two treatments to avoid their failure modes. The following examples illustrate the failure mode of each treatment and motivate the combination of defense.

**Failure Mode 1.** Figure 1a shows an example where the adversary may exploit a dimension by inserting outliers where the benign values have a consistent sign. The outliers can mislead the average toward the non-invariant direction. The robust mean estimator (treatment 2) can trim the outliers

and accurately estimate the mean. In contrast, treatment 1 can fail with a high sign consistency. Because it either lets the highly consistent dimension pass or has to be over-aggressive in zeroing out the dimensions, hurting the model's accuracy on benign data.

**Failure Mode 2.** Figure 1b shows some values with inconsistent signs, which treatment 1 can handle. However, this example can fail the robust mean estimator (treatment 2), whose result has the same sign as the malicious values.

The following sections shall detail the two treatments and discuss their complementary relationship.

## 4.2 AND-MASK

The AND-mask (Parascandolo et al., 2021) computes a dimension-wise mask by inspecting the sign consistency of each dimension across clients. For dimension $k$, the sign consistency is: $|\frac{1}{N} \sum_{i=1}^{N} \text{sign}(\boldsymbol{g}_{i,k})|$. If the sign consistency is below a given threshold $\tau$, the mask element $m_k$ is set to 0, otherwise, $m_k$ is set to 1. The mask along dimension $k$ is defined as:

**Definition 1.** (AND-Mask) For the $k^{\text{th}}$ dimension in the gradient vector, the corresponding mask $m_k$ is defined as:

$$m_k := \mathbf{1}\left[|\sum_{i=1}^{N} \text{sign}(\boldsymbol{g}_{i,k})| \geq \tau\right]. \tag{3}$$

Our defense then multiplies the mask $m$ with the aggregated gradient $\bar{\boldsymbol{g}}$ element-wise, setting the inconsistent dimension to zero.

## 4.3 TRIMMED-MEAN

To complement the AND-mask, our defense broadcasts the trimmed-mean estimator to each gradient dimension. The trimmed-mean estimator alleviates the outlier issue by removing the subset of largest and smallest elements before computing the mean. The largest and smallest elements appear on the two tails of a sorted sequence. Next, we define order statistics and the trimmed mean estimator.

**Definition 2.** (Order Statistics) (Xie et al., 2020b) By sorting the scalar sequence $\{x_i : i \in \{1, ..., N\}, x_i \in \mathbb{R}\}$, we get $x_{1:N} \leq x_{2:N} \leq ... \leq x_{N:N}$, where $x_{i:N}$ is the $i^{\text{th}}$ smallest element in $\{x_i : i \in \{1, ..., N\}\}$.

Then, the trimmed-mean estimator removes a $\alpha \times N$ elements from each tail of the sorted sequence.

**Definition 3.** (Trimmed Mean Estimator) (Xie et al., 2020b) For $\alpha \in [0, 1]$, the $\alpha$-trimmed mean of the set of scalars $x_{i:N} \in \{1, ..., N\}$ is defined as follows:

$$\text{TrMean}(\{x_1, ..., x_N\}; \alpha) = \frac{1}{N - 2 \cdot \lceil \alpha \cdot N \rceil} \sum_{i=\lceil \alpha \cdot N \rceil + 1}^{N - \lceil \alpha \cdot N \rceil} x_{i:N}, \tag{4}$$

where $\lceil . \rceil$ denotes the ceiling function.

## 4.4 OUR APPROACH: INVARIANT AGGREGATOR

Algorithm 1 outlines the steps of our server-side defense that perform aggregation of invariant updates from the clients. The solution is composed of the AND-mask (treatment 1) and trimmed-mean estimator (treatment 2). Our defense applies the two components separately based on the sign consistency of each dimension with a threshold $\tau$.

We show how these two components, i.e., the AND-mask and the trimmed-mean estimator, complement each other. Our analysis considers a single dimension and starts with the robustness of the trimmed mean estimator, which improves the robustness of the AND-mask against outliers. The following theorem extends the robustness guarantee of a modified trimmed-mean estimator (Lugosi & Mendelson, 2021), which is shown in Appendix B, to the conventional trimmed-mean estimator (Definition 3).

**Algorithm 1** Server-side Defense

---

**Input:**
    A set of reported gradients, $\{\boldsymbol{g}_i \mid i \in \{1, ..., N\}\}$;
    Hyper-parameters $\tau, \alpha$;
**Aggregator:**
 1: Compute the AND-mask $m := \mathbf{1}\left[|\sum_{i=1}^{N} \text{sign}(\boldsymbol{g}_i)| \geq \tau\right]$ following Definition 1;
 2: Compute the trimmed-mean $\bar{g} := \text{TrMean}(\{\boldsymbol{g}_1, ..., \boldsymbol{g}_N\}; \alpha)$ under Definition 3;
 3: **return** $m \odot \bar{g}$;

---

**Theorem 4.** *With the trimmed-mean estimator in Definition 3, for a given set of samples $x_1, ..., x_N$, with a corruption level $\eta = \frac{N'}{N}$ and a confidence level $\delta$, set the trim level $\alpha = 8\eta + 12\frac{\log(\frac{4}{\delta})}{N}$, let $a = x_{\alpha N:N}$ and $b = x_{N-\alpha N:N}$ following Definition 2, x be a random variable with variance $\sigma$ and $\bar{x}$ be the estimated mean, with probability at least $\sum_{i=N-\alpha N}^{N-N'} \binom{N-N'}{i} 0.99^i 0.01^{N-N'-i} c^{-4}(1 - 4e^{\frac{-\alpha N}{12}})$, we have:*

$$|\bar{x} - \mathbb{E}[x]| \leq (20\alpha + 10\sqrt{\alpha} + 2c)\sigma \tag{5}$$

The proof is in Appendix B. Theorem 4 bounds the estimation error of a trimmed-mean estimator, which can increase as the variable's variance increases. Multiple factors can increase the variance, such as non-i.i.d. federated data distribution and the stochastic gradient estimation process. In practice, a threat analysis is necessary to specify the maximum number of malicious clients to be tolerated, when the number of malicious clients is unknown. Our goal is to prevent the outliers from misleading the sign. Therefore, the estimation error to expectation ratio, $\frac{|\bar{x}-\mathbb{E}[x]|}{\mathbb{E}[x]}$, is particularly relevant. Since the estimator error for a given $\alpha$ depends on the variance, a high expectation-variance ratio, $\frac{\mathbb{E}[x]}{\sigma}$, is desirable. Then, we show that AND-mask identifies the elements with high expectation-variance ratios, avoiding the robustness degradation of the trimmed-mean estimator. The following theorem suggests that dimensions with a higher expectation-variance ratio have high probabilities of passing AND-mask, and increasing the mask threshold $\tau$ increases the chance of filtering out dimensions with low expectation-variance ratios.

**Theorem 5.** *Given a non-zero expectation-variance ratio $\phi = \frac{\mathbb{E}[x]}{\sigma}$, $N'$ malicious elements, with probability at most $\sum_{i=N-2N'-\tau N}^{\min(N, N-2N'+\tau N)} \binom{N-N'}{i} \phi^{-2i}$, the sign consistency is below $\tau$.*

Appendix B provides the proof. For the $\phi = 0$ case, we may use the probability of the estimated gradient sign being the same as the malicious gradient to replace $\phi^{-2}$. We do not propose directly using the sample mean-variance ratio due to a potential issue: using the sample mean-variance ratio can be over-aggressive when the benign value has a consistent sign but varying magnitudes. Our ablation study in Appendix D.2 shows that AND-mask can preserve more utility than the sample mean-variance ratio when combined with the trimmed-mean estimator.

## 5 ROBUSTNESS ANALYSIS FOR A SIMPLIFIED MODEL

To further motivate our approach, we consider a more direct robustness analysis for a specific generative model. This analysis discusses how the features impact the gradient sign consistency, when the two failure modes in Section 4.1 appear, and why we need a combination of defenses guaranteed by Theorems 4 and 5. First, we outline some preliminaries useful for this analysis.

### 5.1 PRELIMINARIES

We consider a binary prediction task $y \in \{0, 1\}$ with a linear model $h(\boldsymbol{x}) = \boldsymbol{w}^\top \boldsymbol{x}$ and a decision rule $\hat{y} = \mathbf{1}_{\boldsymbol{w}^\top \boldsymbol{x} \geq 0}(\boldsymbol{w}^\top \boldsymbol{x})$. The training procedure uses a Sigmoid activation function $s(z) = \frac{1}{1+e^{-z}}$ and a logistic loss function $\ell(z, y) = -y \cdot \log(z) - (1 - y) \cdot \log(1 - z)$, where $z = h(\boldsymbol{x})$.

**Data Model.** We assume that the samples per class are balanced on benign clients. The data samples come from a non-i.i.d. Gaussian distribution with a diagonal covariance matrix. On the $i^{\text{th}}$ client, for the $k^{\text{th}}$ feature, we have:

$$\boldsymbol{x}_k \sim \mathcal{N}\Big((2y - 1) \cdot \boldsymbol{\mu}_{i,k}, \boldsymbol{\sigma}_{i,k}\Big). \tag{6}$$

**Backdoor Attack.** We consider a single feature backdoor attack where the trigger is the $k^{\text{th}}$ feature $\boldsymbol{x}_k$. A backdoor using a specific feature is common (Wang et al., 2020). To backdoor images, the adversary often selects a pixel pattern or semantic pattern (e.g., blue color on airplanes). For text data, the trigger could be a dedicated set of characters . The malicious clients attack in collusion using the same backdoor samples, meaning that $\boldsymbol{\mu}_{i,k} = \boldsymbol{\mu}_{i',k} \neq 0, \forall i, i' \in \{N - N' + 1, N\}$. To simplify the notation, we omit the subscript $i$ for simplicity while focusing on a single client. Since the trigger is not useful or does not appear on benign clients, we assume $\boldsymbol{\mu}_{i,k} = 0, \forall i \in \{1, N - N'\}$

**Objective.** The backdoor attack is effective if the model entangles the trigger-label correlation. For a non-zero $\boldsymbol{\mu}_k$ of the trigger $\boldsymbol{x}_k$, $\boldsymbol{w}_k\boldsymbol{\mu}_k > 0$ is a necessary condition for a model to entangle the trigger-label correlation. Therefore, in the case when $\boldsymbol{\mu}_k > 0$, avoiding $\boldsymbol{w}_k$ from increasing and enforcing $\boldsymbol{w}_k$ to decrease while $\boldsymbol{w}_k > 0$ can mitigate the backdoor attack.

## 5.2 CONNECTING FEATURES TO GRADIENTS

We define the invariance of a feature by measuring its feature-label correlation (e.g., positive or negative) consistency across clients.

**Definition 6.** For a given feature $k$, its invariance $p$ is defined as: $p = \left| \frac{1}{N} \cdot \sum_{i=1}^{N} \text{sign}(\boldsymbol{\mu}_{i,k}) \right|$.

Similarly, we define the sign consistency of a gradient dimension $k$:

**Definition 7.** With a linear model $h(x) = \boldsymbol{w}^\top \boldsymbol{x}$, a Sigmoid activation function $s$, and $N$ clients, a $q$-consistent gradient w.r.t. $\boldsymbol{w}_k$ satisfies: $\left| \frac{1}{N} \cdot \sum_{i=1}^{N} \text{sign}\left( \mathbb{E}_{\boldsymbol{x},y \sim \mathcal{D}_i} \left[ \nabla_{\boldsymbol{w}_k} \ell(s(\boldsymbol{w}^\top \boldsymbol{x}), y) \right] \right) \right| = q$.

Under Definitions 6 and 7, we discuss the connection between feature invariance and dimension-wise gradient sign consistency. First, we need to analyze the behavior of the gradient sign per client.

**Theorem 8.** *For a linear model with a Sigmoid activation function $s$ and the logistic loss $\ell$, under our Gaussian data model, on the $k^{\text{th}}$ feature with a non-zero $\boldsymbol{\mu}_k$, if $\boldsymbol{w}_k\boldsymbol{\mu}_k \leq 0$, we have* $\text{sign}\left( \mathbb{E}_{\boldsymbol{x},y \sim \mathcal{D}_i} \left[ \nabla_{\boldsymbol{w}_k} \ell(s(\boldsymbol{w}^\top \cdot \boldsymbol{x}), y) \right] \right) = \text{sign}(\boldsymbol{\mu}_k)$. *In addition, if $\boldsymbol{\mu}_k = 0$, we have* $\text{sign}\left( \mathbb{E}_{\boldsymbol{x},y \sim \mathcal{D}_i} \left[ \nabla_{\boldsymbol{w}_k} \ell(s(\boldsymbol{w}^\top \cdot \boldsymbol{x}), y) \right] \right) = \text{sign}(\boldsymbol{w}_k)$.

The proof is provided in Appendix B. The result in Theorem 8 is intuitive. With a non-zero $\boldsymbol{\mu}_k$, if $\text{sign}(\boldsymbol{w}_k)$ agrees with $\text{sign}(\boldsymbol{\mu}_k)$, the gradient sign can be indefinite because the weight $\boldsymbol{w}_k$ can be either larger or smaller than the optimal $\boldsymbol{w}_k^*$. Otherwise, the gradient has the same sign as $\boldsymbol{\mu}_k$. If $\boldsymbol{\mu}_k = 0$, $\boldsymbol{w}_k$ shall shrink to 0. Then, we outline the conditions that lead to inconsistent signs.

**Corollary 9.** *Under Theorem 8, suppose $\boldsymbol{x}_k$ represents the trigger, $\boldsymbol{\mu}_k = 0$ on the $N - N'$ benign clients and the $N'$ malicious clients share the same non-zero $\boldsymbol{\mu}_k$, if $\boldsymbol{w}_k = 0$, the gradient w.r.t. $\boldsymbol{w}_k$ is $\frac{N'}{N}$-consistent. In addition, if $\boldsymbol{w}_k\boldsymbol{\mu}_k > 0$, the gradient w.r.t. $\boldsymbol{w}_k$ is at least $(1 - \frac{N'}{N})$-consistent. The gradient w.r.t. $\boldsymbol{w}_k$ is 1-consistent if $\boldsymbol{w}_k\boldsymbol{\mu}_k < 0$.*

Counting the gradient signs yields the result. Corollary 9 suggests that if $\boldsymbol{w}_k\boldsymbol{\mu}_k > 0$ and the model entangles the backdoor, the expected gradient w.r.t. $\boldsymbol{w}_k$ can align with $\boldsymbol{\mu}_k$ on malicious clients and conflict with $\boldsymbol{\mu}_k$ on benign clients. Then, employing the trimmed-mean estimator to remove the malicious values can recover the invariant direction pointed by benign clients (Theorem 4) and thereby shrink $\boldsymbol{\mu}_k$. If $\boldsymbol{\mu}_k = 0$, using AND-mask can mask out the gradients w.r.t. $\boldsymbol{w}_k$ from the malicious clients (Theorem 5). If $\boldsymbol{w}_k$ remains 0, the model parameterized by $\boldsymbol{w}$ is robust to the trigger on $\boldsymbol{x}_k$. It is worth noting that the gradients w.r.t. $\boldsymbol{w}_k$ is 1-consistent and align to $\boldsymbol{\mu}_k$ when $\boldsymbol{w}_k\boldsymbol{\mu}_k < 0$. Such a consistent gradient may overshoot and flip the sign of $\boldsymbol{w}_k$. Reducing the learning rate can alleviate overshooting and the trimmed-mean estimator will guarantee $\boldsymbol{w}_k$ to shrink after the overshooting.

**Connection to Failure Modes.** If $\boldsymbol{w}_k\boldsymbol{\mu}_k > 0$, the gradients w.r.t. $\boldsymbol{w}_k$ have a consistent but non-zero expectation among benign clients, causing the failure mode 1 in Section 4.1. On the other hand, the gradient variance can diversify the estimated gradient sign and cause the failure mode 2.

Appendix D.1 further provides simulation results under the linear regime (Section 5.1), showing that the AND-mask can prevent the adversary from exploiting $\boldsymbol{w}_k$ and the trimmed-mean estimator helps shrink $\boldsymbol{w}_k$.

# 6 EXPERIMENTS

We evaluate our defense on three realistic tasks on three different data types: (1) object recognition with visual data, (2) sentiment analysis with text data, and (3) phishing email detection with tabular data. We employ the state-of-the-art edge-case backdoor attack (Wang et al., 2020) to generate backdoor samples and evaluate of defense and existing defenses against it.

**Additional Results.** The simulations, an ablation study, an evaluation of the hyper-parameter sensitivity, evaluations with additional attack strategies, and empirical verifications of the two failure modes can be found in the Appendix D.

## 6.1 EXPERIMENTAL SETUP

We briefly summarize our setup and report more details in Appendix C.

**Metrics.** Our experiments employ two metrics: the main task accuracy ($Acc_M$) estimated on the benign samples and the backdoor task accuracy ($Acc_B$) over backdoor samples. A defense is designed to reduce the model's accuracy on backdoor task and maintain the utility on the main task.

**Datasets.** The visual data of the object detection task and text data of the sentiment analysis task are from CIFAR-10 (Krizhevsky, 2009; McMahan et al., 2017) and Twitter (Caldas et al., 2018), respectively. Each phishing email data sample has 45 standardized numerical features of the sender that represent the sender reputation scores. A large reputation score may indicate a phishing email. The reputation scores come from peer-reviewers in a reputation system (Jøsang et al., 2007). The adversary may use malicious clients to manipulate the reputation.

**Federated Learning Setup.** We consider horizontal federated learning (Kairouz et al., 2021) where the clients share the same feature and label spaces. The number of clients are 100 for the three tasks. The server sample 20 clients at each round on the CIFAR-10 and phishing email experiments. We reduce the sampled client number to 15 on the Twitter experiment due to limited hardware memory.

**Backdoor Attack Setup.** The adversary employs the edge-case backdoor attack, where it selects the data samples with low marginal probability in their data distribution to create backdoor samples. The visual and text backdoor samples follow the previous work (Wang et al., 2020). For the tabular data, we select the $38^{th}$ feature (reputation), whose value is 0 on most of the data samples. Then, we let the adversary manipulate the $38^{th}$ feature to 0.2 that has a low probability density on phishing emails and flip the label to non-phishing.

The adversary can control 20% clients on the CIFAR-10 and phishing email experiments and 10% clients on the Twitter experiment. Section 6.2 explains the different experiment configurations. Such an adversary is considered strong in practice (Shejwalkar et al., 2022). We consider a strong adversary because defending against strong adversary yields robustness against weak adversary, whose effectiveness is already shown (Wang et al., 2020). The adversary only uses backdoor samples during training.

## 6.2 RESULT AND COMPARISON TO PRIOR WORKS

**Our results.** Table 1 summarizes the performance of each defense on three tasks. Our approach decreases the backdoor task accuracy by 61.6% on average. The edge-case backdoor attack on the text sentiment analysis task (Twitter) is more difficult to defend and our approach mitigates the accuracy increase on the backdoor task by 41.7%. We hypothesize that the text sentiment analysis task has few invariant and benign features. For example, the shape features (Sun et al., 2021) in object classification tasks can be invariant across objects. In contrast, the sentiment largely depends on the entire sentence instead of a few symbols or features. Then, we discuss the limitations of prior defenses.

**Vector-wise.** Common vector-wise defenses such as Krum estimate pair-wise similarities in terms of Euclidean distance (Blanchard et al., 2017) (Krum and multi-Krum) of cosime similarity (Nguyen et al., 2021) (multi-Krum$_C$) between each gradient and others. The gradients that are dissimilar to others are removed. The vector-wise view is insufficient for defending against backdoor attacks because backdoor attacks can succeed by manipulating a tiny subset of parameters (e.g. 5%) (Wu & Wang, 2021) without incurring much vector-wise difference. In practice, we observe that the malicious gradients can get high similarity scores and circumvent vector-wise defenses.

Table 1: Accuracy of Aggregators under Edge-case Backdoor Attack. Our approach reduces the model accuracy on backdoor samples by 61.7% on average, mitigating the backdoor attack, and achieves a comparable utility on benign samples as the standard FedAvg aggregator.

| Method | CIFAR-10 | | Twitter | | Phishing | |
|---|---|---|---|---|---|---|
| | $Acc_M$ | $Acc_B$ | $Acc_M$ | $Acc_B$ | $Acc_M$ | $Acc_B$ |
| FedAvg | $.679 \pm .001$ | $.717 \pm .001$ | $.722 \pm .001$ | $.440 \pm .001$ | $.999 \pm .001$ | $.999 \pm .001$ |
| Krum | $.140 \pm .001$ | $.275 \pm .012$ | $.579 \pm .001$ | $.766 \pm .002$ | $.999 \pm .001$ | $.999 \pm .001$ |
| Multi-Krum | $.541 \pm .002$ | $.923 \pm .021$ | $.727 \pm .001$ | $.656 \pm .008$ | $.999 \pm .001$ | $.999 \pm .001$ |
| Multi-Krum$_C$ | $.681 \pm .002$ | $.821 \pm .001$ | $.594 \pm .002$ | $.701 \pm .001$ | $.999 \pm .001$ | $.333 \pm .333$ |
| Trimmed-Mean | $.687 \pm .001$ | $.512 \pm .002$ | $.728 \pm .001$ | $.640 \pm .016$ | $.999 \pm .001$ | $.999 \pm .001$ |
| Krum Trimmed-Mean | $.682 \pm .001$ | $.607 \pm .002$ | $.727 \pm .001$ | $.641 \pm .001$ | $.999 \pm .001$ | $.999 \pm .001$ |
| Sign-SGD | $.301 \pm .005$ | $\mathbf{.000} \pm \mathbf{.001}$ | $.610 \pm .003$ | $.751 \pm .076$ | $.999 \pm .000$ | $.667 \pm .333$ |
| Weak-DP | $.454 \pm .003$ | $.828 \pm .003$ | $.667 \pm .001$ | $.374 \pm .002$ | $.999 \pm .001$ | $.999 \pm .001$ |
| Freezing Layers | $.415 \pm .001$ | $.572 \pm .002$ | N\A | N\A | N\A | N\A |
| FoolsGold | $.667 \pm .001$ | $.109 \pm .001$ | $.726 \pm .002$ | $.357 \pm .001$ | $.999 \pm .001$ | $.999 \pm .001$ |
| RFA | $.685 \pm .001$ | $.853 \pm .002$ | $.718 \pm .001$ | $.704 \pm .002$ | $.999 \pm .001$ | $.999 \pm .001$ |
| SparseFed | $.662 \pm .001$ | $.984 \pm .001$ | $.667 \pm .001$ | $.608 \pm .002$ | $.999 \pm .001$ | $.999 \pm .001$ |
| No Attack | $.718 \pm .001$ | $.000 \pm .001$ | $.731 \pm .001$ | $.095 \pm .001$ | $.999 \pm .001$ | $.000 \pm .001$ |
| **Ours** | $.677 \pm .001$ | $\mathbf{.001} \pm \mathbf{.001}$ | $.687 \pm .001$ | $\mathbf{.296} \pm \mathbf{.003}$ | $.999 \pm .001$ | $\mathbf{.000} \pm \mathbf{.001}$ |

Note: The numbers are average accuracy over three runs. Variance is rounded up.

**Dimension-wise.** Failure mode 2 in Section 4.1 shows the limitation of the trimmed-mean estimator, which was the most effective defense against the edge-case backdoor attack. We also include Sign-SGD with majority vote (Bernstein et al., 2019) as a defense, which binarizes the gradient and takes the majority vote as the aggregation result. However, Sign-SGD struggles to train a large federated model (e.g., Resnet-18 on CIFAR-10) and can suffer from failure mode 1 where the clients have diverse signs. Then, the adversary can put more weight on one side and mislead the voting result.

**Combination.** A naive combination of multi-Krum and the trimmed-mean estimator fails to defend against the backdoor attack because neither multi-Krum nor the trimmed-mean estimator avoids the failure mode of the other.

**Weak-DP.** The weak-DP defense (Sun et al., 2019) first bounds the gradient norms, then add additive noise (e.g., Gaussian noise) to the gradient vector. The edge-case backdoor attack can work without scaling up the gradients, circumventing the norm bounding. For the additive noise, we hypothesize that in some dimensions, the difference between malicious and benign gradients can be too large for the Gaussian noise to blur their boundary.

**Freezing Layers.** Since we employ a pre-trained Resnet-18 (He et al., 2016) on CIFAR-10, freezing the convolution layers may avoid entangling the trigger. However, this approach lacks empirical robustness, possibly because the adversary can use semantic features (e.g., blue color on airplanes) that the pre-trained model already learns as triggers.

**Advanced Defenses.** FoolsGold (Fung et al., 2020) down-weights an update if that update has a high cosine similarity with another update. There are many ways to diversify updates. Malicious clients may leverage the stochastic gradient estimation process or mix backdoor samples with benign samples, whose distribution can differ across clients. RFA (Pillutla et al., 2022) computes geometric medians as the aggregation result, which is shown to be ineffective (Wang et al., 2020). SparseFed (Panda et al., 2022) only accepts elements with large magnitude in the aggregation results. However, benign and malicious updates can contribute to large magnitudes.

## 7 CONCLUSION AND FUTURE WORK

This paper shows how to defend against backdoor attacks by focusing on the invariant directions in the model optimization trajectory. Enforcing the model to follow the invariant direction requires AND-mask to compute the sign-consistency of each gradient dimension, which estimates how invariant a dimension-wise direction can be, and use the trimmed-mean estimator to guarantee the model follows the invariant direction within each dimension. Both theoretical and empirical results demonstrate the combination of AND-mask and the trimmed-mean estimator is necessary and effective. Further defending against more advanced backdoor attacks such as invisible backdoors that add calibrated noise to all the features (Li et al., 2021b; Manoj & Blum, 2021) can be interesting.

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
