# OpenReview forum: "Invariant Aggregator for Defending against Federated Backdoor Attacks"
_ICLR.cc/2023/Conference — Submitted to ICLR 2023_

### Official Review · Reviewer_9BAU · 2022-10-20

**Confidence:** 5
**Correctness:** 3
**Technical Novelty And Significance:** 3
**Empirical Novelty And Significance:** 2
**Recommendation:** 5

**Clarity, Quality, Novelty And Reproducibility:**

The paper is easy to follow in general. The paper novelty is moderate--It shows new theoretical results but are narrow. The experimental setup is clear, and I think the results can be reproducible.

**Strength And Weaknesses:**

Strengths
+ The studied problem is important
+ Detailed analysis on why AND-mask and Trimmed mean together to enhance robustness

Weaknesses
-Some details are unclear
-Theoretical results are narrow
-Lack of comparison with state-of-the-art defenses

**Summary Of The Paper:**

The paper studies defenses against federated learning backdoor attacks. The paper proposes to force the model optimization trajectory to focus on the invariant directions that are generally useful for utility and avoid selecting directions that favor malicious clients. Particularly, the authors consider the consistency of the client model update as an estimation of the invariance, where the AND-mask and Trimmed mean are used together to measure the sign consistency. The proposed robust aggregator is evaluated on several datasets to defend against backdoor attacks, and show a promising utility-robustness tradeoff.


**Summary Of The Review:**

Detailed comments:

I do not quite understand Figure 1(a): When using the AND-mask, does the result show negative signs, which returns the accurate sign of the benign dimensions?

What does expectation ratio indicate in Theorem 4? What is the rough error bound in Equation (5)? Is it too large? Similarly, what is the rough probability in Theorem 5?

Theorem 4 & 5 only relate to a single feature dimension? Can the result be generalized to the feature vector? What is the error bound?

Theorem 8 is for the simple linear model. I am not sure whether the theoretical results can be applicable to general nonlinear model.

I am also curious about the attack scenario that can break both the AND-mask and trimmed-mean.

Can the proposed solution be generalized to defend against model poisoning attacks?


Which features are filtered by AND-mask and which filtered by the Trimmed mean? I would like to see whether they are complementary or have a large overlap.

Why no attacks in Table 1 has non-zero backdoor accuracy?

---

> ### Author Response · Authors · 2022-11-17
> **Response to Reviewer 9BAU (2/2)**
>
> **I am also curious about the attack scenario that can break both the AND-mask and trimmed-mean.**
>
> We are not aware of such an attack scenario to our best knowledge. To further evaluate the robustness of our approach, we consider an adaptive adversary to exploit invariant directions. Invariant directions are preferred by our approach.
>
> Specifically, we consider adaptive adversaries that zero the dimensions where signs of malicious updates conflict with the majority signs of benign updates. Then, adversaries scale up the remaining elements such that update magnitudes remain the same. Additional experimental result in Appendix D.4 of the post-rebuttal version shows that such an adaptive strategy only increases the accuracy on backdoor samples by 3% on the Twitter dataset and is ineffective on the other two datasets, compared to the non-adaptive attacks.
>
> &nbsp;
>
> **Can the proposed solution be generalized to defend against model poisoning attacks?**
>
> Our robustness guarantees (Theorems 4&5) are generally applicable, although we focus on the threat model of backdoor attacks (Sections 3 and 5.1) in this work. Empirically evaluating our approach against untargeted poisoning attacks can be interesting for future works.
>
> &nbsp;
>
> **Which features are filtered by AND-mask and which filtered by the Trimmed mean? I would like to see whether they are complementary or have a large overlap.**
>
> Only the AND-mask filters out invariant features by setting inconsistent dimensions to 0. The trimmed-mean estimator protects the remaining consistent dimensions and avoids outliers from malicious clients to mislead aggregation results (Figure 1(a)).
>
> &nbsp;
>
> **Why no attacks in Table 1 has non-zero backdoor accuracy?**
>
> That’s a good point. Even without attacks, models can err on edge-case (rare) examples, especially on difficult tasks such as text sentiment prediction.
>
> &nbsp;
>
> **Theoretical results are narrow.**
>
> Our robustness guarantees in Theorems 4 and 5 are widely applicable. In addition, the simplicity of our defense and the strength of the empirical results suggest wide applicability.
>
> &nbsp;
>
> **Lack of comparison with state-of-the-art defenses.**
>
> We have compared our approach with FoolsGold [3], RFA [4], and SparseFed [5], all state-of-the-art in this area. Results in Section 6.2 of the post-rebuttal version show that our approach is more robust than the recent defenses by 6% - 99%.
>
> &nbsp;
>
> **Reference**
>
> [1] Lugosi, Gabor, and Shahar Mendelson. "Robust multivariate mean estimation: the optimality of trimmed mean." The Annals of Statistics 49.1 (2021): 393-410.
>
> [2] Manoj, Naren, and Avrim Blum. "Excess capacity and backdoor poisoning." Advances in Neural Information Processing Systems 34 (2021): 20373-20384.
>
> [3] Fung, Clement, Chris JM Yoon, and Ivan Beschastnikh. "The limitations of federated learning in sybil settings." 23rd International Symposium on Research in Attacks, Intrusions and Defenses (RAID 2020). 2020.
>
> [4] Pillutla, Krishna, Sham M. Kakade, and Zaid Harchaoui. "Robust aggregation for federated learning." IEEE Transactions on Signal Processing 70 (2022): 1142-1154.
>
> [5] Panda, Ashwinee, et al. "SparseFed: Mitigating Model Poisoning Attacks in Federated Learning with Sparsification." International Conference on Artificial Intelligence and Statistics. PMLR, 2022.

---

> ### Author Response · Authors · 2022-11-17
> **Response to Reviewer 9BAU (1/2)**
>
> Thanks for the comments. We have clarified Figure 1(a) and provided more details of our theoretical analysis in the response. We have also added three state-of-the-art defenses to Section 6.2 of the post-rebuttal version.
>
> &nbsp;
>
> **I do not quite understand Figure 1(a): When using the AND-mask, does the result show negative signs, which returns the accurate sign of the benign dimensions?**
>
> In figure 1(a), three benign clients vote for the negative sign, while a malicious client votes for the positive sign. This dimension is consistent because 75% of clients agree on the negative sign. However, the sign of the average can be positive due to the malicious client, differing from the negative sign that is voted by benign clients.
>
> &nbsp;
>
> **What does expectation ratio indicate in Theorem 4\?**
>
> The expectation-variance ratio $\frac{\mathbb{E}[x]}{\sigma}$ indicates the estimation error-expectation ratio $\frac{|\bar{x} - \mathbb{E}[x]|}{\mathbb{E}[x]}$. Here, the estimation error is the absolute difference between the estimated mean and the expectation. The estimation error-expectation ratio is important because if the ratio is below 1, the estimated mean has the same sign as the expectation. To see this relationship, we could divide both sides in Equation (5) by the expectation $\mathbb{E}[x]$. We have improved Theorem 4 in the post-rebuttal version to clarify the relationship between the estimation error and the variance.
>
> &nbsp;
>
> **What is the rough error bound in Equation (5)? Is it too large\?**
>
> Thanks for raising the question. In the post-rebuttal version, the error bound in Equation (5) consists of the variance $\sigma$ and a few constants: trim ratio $\alpha$, confidence level $c$. Suppose there are 200 clients in total, with 8 malicious clients. Assume a variable $x$ has an expectation 1.0 and a variance 0.05. Let $c$ = 1. With a probability at least 0.845, the estimation error is below 0.754, less than the expectation. Such an error bound has a similar magnitude to the bound in a recent work [1], which bounds the estimation error below 0.301 with a probability at least 0.989.
>
> &nbsp;
>
> **Similarly, what is the rough probability in Theorem 5?**
>
> Suppose there are 30 clients in total, with 6 malicious clients. If the expectation-variance ratio of the $i^\mathrm{th}$ dimension is 2.0, the AND-mask rejects the $i^\mathrm{th}$ dimension with a probability of 0.207. If the expectation-variance ratio is 3.0, the probability of rejection decreases to 1e-5. These numerical results suggest that AND-mask rejects dimensions with low expectation-variance ratios.
>
> &nbsp;
>
> **Theorem 4 & 5 only relate to a single feature dimension? Can the result be generalized to the feature vector? What is the error bound?**
>
> Theorems 4 & 5 do not assume linear models and apply to a single dimension of gradient vectors. Our dimension-wise analysis is consistent with recent works on trimmed-mean estimators [1].
>
> Summing up the dimension-wise error bounds and taking the product of dimension-wise probabilities generalize our analysis to multi-dimension settings and obtain vector-wise error bounds.
>
> &nbsp;
>
> **Theorem 8 is for the simple linear model. I am not sure whether the theoretical results can be applicable to general nonlinear model.**
>
> Probing model behaviors against backdoor attacks using a linear model is common [2]. Since the goal of Theorem 8 and Corollary 9 is to show sign conflictions exist and cause the two failure modes, we resort to empirical evaluation of sign conflictions and failure modes with general nonlinear models in Appendix D.5 of the post-rebuttal version. Our results show that the sign conflictions exist, and our approach avoids the failure modes caused by the sign confliction.

---

### Official Review · Reviewer_f8s8 · 2022-10-24

**Confidence:** 4
**Correctness:** 2
**Technical Novelty And Significance:** 2
**Empirical Novelty And Significance:** 2
**Recommendation:** 3

**Clarity, Quality, Novelty And Reproducibility:**

### Clarity

The paper is mostly clear and easy to follow. It misses the details of baseline methods in the evaluation.


### Quality and Novelty

The novelty is limited and the motivation needs further validation. The paper fails to evaluate more and stronger backdoor attacks and compare with state-of-the-art defense techniques.


### Reproducibility

The submission does not include the implementation and the details of baselines are missing.


**Strength And Weaknesses:**

### Strength

+ Important problem of defending backdoor attacks in federated learning
+ Easy to follow in most places


### Weaknesses

- No empirical validation of the motivation
- Limited novelty
- No evaluation on more existing backdoor attacks
- No evaluation on adaptive attacks
- Missing state-of-the-art defense baselines
- Missing references and details of baseline techniques

### Detailed comments

Defending backdoor attacks in federated learning is a timely and important topic. This paper aims to address this problem by leveraging the combination of existing techniques. The paper is mostly easy to follows. There are a few aspects not well addressed in the paper.

* In Figure 1, the paper uses two illustrative examples to demonstrate the failure scenarios of the two existing methods respectively. However, there is no empirical result showing that such cases exist in real backdoor attacks. In addition, there could be other attack scenarios. For example, both benign and malicious values are on the same side of the sign. But the malicious value can be far from the benign ones. Other cases may also exist. It shall be empirically studied and validated regarding those cases.

* The novelty is limited. This paper simply combines the two existing techniques, without any modifications. The combination is completely based on the two illustrative examples given in Figure 1, which could be far from comprehensive. The theoretical analysis is conducted on a linear classifier, which is also based on the two assumed attack scenarios. The analysis on the gradient sign does not address the aforementioned case where the malicious gradient shares the same sign as benign ones.

* There are many other backdoor attacks in federated learning such as [1-2]. Particularly, there is an attack setting in [2] where the attacker is selected in each round and continuously participates in the training from beginning to the end, which is a much more aggressive attack. The proposed method shall be evaluated on those backdoor attacks.

* Since the proposed defense is based on the assumption of the malicious gradient sign deviate from benign ones, an adaptive attacker can match the malicious gradient sign with a benign one during the attack. Will the proposed defense still be effective?

* There are other state-of-the-art defense techniques [3-5]. They shall be compared with in the evaluation.

* There are no references and detailed descriptions of those baselines in Table 1.


### References

[1] Bagdasaryan, Eugene, et al. "How to backdoor federated learning." AISTATS 2020.

[2] Xie, Chulin, et al. "Dba: Distributed backdoor attacks against federated learning." ICLR 2019.

[3] Pillutla, Krishna, et al. "Robust aggregation for federated learning." IEEE Transactions on Signal Processing 2022.

[4] Fung, Clement, et al. "The limitations of federated learning in sybil settings." RAID 2020.

[5] Cao, Xiaoyu, et al. "FLTrust: Byzantine-robust Federated Learning via Trust Bootstrapping." NDSS 2021.

**Summary Of The Paper:**

This paper aims to defend backdoor attacks in federated learning. Particularly, it proposes to combine existing two techniques, AND-mask and trimmed-mean estimator, to remove suspicious gradient dimensions. This is based on the assumption that the gradients from benign clients have the same sign while those from malicious clients are different from benign ones. This paper also provides theoretical analysis on the gradient sign consistency and how the combination of the two existing techniques defends backdoor attacks. The evaluation is conducted on three datasets and one backdoor attack. Compared to a few baseline methods, the proposed approach has a better defense performance.

**Summary Of The Review:**

This paper addresses an import problem but lacks novelty and comprehensive empirical results.

---

> ### Author Response · Authors · 2022-11-17
> **Response to Reviewer f8s8 (2/2)**
>
> **There are many other backdoor attacks in federated learning such as [1-2]. Particularly, there is an attack setting in [2] where the attacker is selected in each round and continuously participates in the training from beginning to the end, which is a much more aggressive attack. The proposed method shall be evaluated on those backdoor attacks.**
>
> We considered a fixed-pool setting in the experiments, where attackers can be selected from client pools during client sampling in each round and continuously participates in the training from the beginning to the end. We will clarify this in the updated version.
>
> Our experiments included two backdoor attacks because model replacement and train-and-scale strategies from [2] are also considered in the edge-case backdoor framework (Section 2 of [3]). For the train-and-scale strategy, we scaled malicious updates to the norm clipping threshold. We don’t assume additional anomaly detectors, so the train-and-scale strategy, which is particularly designed to circumvent anomaly detectors, is not considered.
>
> In addition, we evaluated our approach with three types of triggers: the semantic trigger on CIFAR-10, word trigger on Twitter, and value trigger on Phishing.
>
> In the post-rebuttal version (Appendix D.4), we have completed additional evaluations of our defense against **4 more backdoor attacks**: distributed backdoor attacks [3], adaptive backdoor attacks that are particularly designed against our defense, colluding backdoor attacks [4], and adaptive colluding backdoor attacks. Our results show that our approach remains effective and loses no more than 5% robustness against the four improved attacks. For example, our approach decreases the average accuracy on CIFAR-10 backdoor samples from 0.717 to 0.001 against edge-case backdoor attacks and decreases the average accuracy on CIFAR-10 backdoor samples from 0.717 to 0.049 against improved backdoor attacks.
>
> &nbsp;
>
> **There are other state-of-the-art defense techniques [3-5]. They shall be compared with in the evaluation.**
>
> We have added the suggested experiments to Section 6.2 of the post-rebuttal version. The defense techniques include FoolsGold [5], RFA [6], and SparseFed [4]. FLTust does not apply to our setting, which does not assume the availability of bootstrapping datasets. Our approach is more robust than the recent defenses by 6% - 99%.
>
> &nbsp;
>
> **There are no references and detailed descriptions of those baselines in Table 1.**
>
> We have baseline descriptions and references in Section 6.2 in the pre-rebuttal version.
>
> &nbsp;
>
> **References**
>
> [1] Manoj, Naren, and Avrim Blum. "Excess capacity and backdoor poisoning." Advances in Neural Information Processing Systems 34 (2021): 20373-20384.
>
> [2] Bagdasaryan, Eugene, et al. "How to backdoor federated learning." International Conference on Artificial Intelligence and Statistics. PMLR, 2020.
>
> [3] Wang, Hongyi, et al. "Attack of the tails: Yes, you really can backdoor federated learning." Advances in Neural Information Processing Systems 33 (2020): 16070-16084.
>
> [4] Xie, Chulin, et al. "Dba: Distributed backdoor attacks against federated learning." International Conference on Learning Representations. 2019.
>
> [5] Panda, Ashwinee, et al. "SparseFed: Mitigating Model Poisoning Attacks in Federated Learning with Sparsification." International Conference on Artificial Intelligence and Statistics. PMLR, 2022.
>
> [5] Fung, Clement, Chris JM Yoon, and Ivan Beschastnikh. "The limitations of federated learning in sybil settings." 23rd International Symposium on Research in Attacks, Intrusions and Defenses (RAID 2020). 2020.
>
> [6] Pillutla, Krishna, Sham M. Kakade, and Zaid Harchaoui. "Robust aggregation for federated learning." IEEE Transactions on Signal Processing 70 (2022): 1142-1154.

---

> ### Author Response · Authors · 2022-11-17
> **Response to Reviewer f8s8 (1/2)**
>
> Thanks for the comments. In the response, we have clarified the novelty of our approach, the applicability of our robustness guarantees, and the attack setup in our experiments. We have also added additional experiments with strong attacks and recent defenses to the post-rebuttal version.
>
> &nbsp;
>
> **In Figure 1, the paper uses two illustrative examples to demonstrate the failure scenarios of the two existing methods respectively. However, there is no empirical result showing that such cases exist in real backdoor attacks.**
>
> We added additional experiments to Appendix D.5 of the post-rebuttal version, showing that the two failure mode exists. The trimmed-mean estimators help failure mode 1 but do not treat failure mode 2. Therefore, we need AND-mask to directly set inconsistent dimensions in failure mode 2 to 0 so malicious updates can not mislead the aggregation results to benefit malicious clients.
>
> &nbsp;
>
> **In addition, there could be other attack scenarios. For example, both benign and malicious values are on the same side of the sign. But the malicious value can be far from the benign ones. Other cases may also exist. It shall be empirically studied and validated regarding those cases. Since the proposed defense is based on the assumption of the malicious gradient sign deviate from benign ones, an adaptive attacker can match the malicious gradient sign with a benign one during the attack. Will the proposed defense still be effective?**
>
> Thanks for the suggested attack. We have conducted an additional ablation study, showing that matching gradient signs does not circumvent our defense (Appendix D.4 of the post-rebuttal version). Specifically, we consider adaptive adversaries that zero the dimensions where signs of malicious updates conflict with the majority signs of benign updates. Then, adversaries scale up the remaining elements such that update magnitudes remain the same. We also include colluding attacks that let malicious clients report the same malicious update to increase the sign consistency between malicious updates.
>
> Our results show that our approach remains effective and loses no more than 5% robustness against improved attacks. For example, our approach decreases the average accuracy on CIFAR-10 backdoor samples from 0.717 to 0.001 against edge-case backdoor attacks and decreases the average accuracy on CIFAR-10 backdoor samples from 0.717 to 0.049 against improved backdoor attacks.
>
> &nbsp;
>
> **The novelty is limited. This paper simply combines the two existing techniques, without any modifications.**
>
> We respectfully disagree! The novelty of our work is two-fold. First, we show that focusing on invariant directions in the model optimization trajectory (using the AND-mask) can help defend against backdoor attacks. Neither the perspective of invariance directions nor the specific implementation using the AND-mask has not yet been explored in previous defenses against federated backdoor attacks.
>
> Second, we analyze the failure mode of the AND-mask and enhance it with the trimmed-mean estimator. We further demonstrate the complementary relationship (Theorems 4 and 5) between the AND-mask and the trimmed-mean estimator. The failure modes and this complementary relationship have not yet been discovered in prior works.
>
> Importantly, Table 1 in Section 6.2 further shows that a straightforward combination of existing approaches (i.e., Krum + trimmed-mean) does not succeed.
>
> &nbsp;
>
> **The combination is completely based on the two illustrative examples given in Figure 1, which could be far from comprehensive.**
>
> We supported the combination of AND-mask and the trimmed-mean estimator with comprehensive analysis in Theorems 4 and 5. Figure 1 provides motivating examples to ease the reading instead of serving as the base of our approach.
>
> &nbsp;
>
> **The theoretical analysis is conducted on a linear classifier, which is also based on the two assumed attack scenarios.**
>
> We would like to remind the reviewer our robustness guarantees (Theorems 4 and 5) do not assume linear models, and using linear models to explore models’ behaviors against backdoor attacks is a standard pragmatic approach (e.g.,  [1]).
>
> &nbsp;
>
> **The analysis on the gradient sign does not address the aforementioned case where the malicious gradient shares the same sign as benign ones.**
>
> We would like to remind the reviewer that the analysis on the gradient sign shows when the two failure modes happen instead of providing robustness guarantees. The robustness guarantees of our approach (Theorems 4 and 5) apply to the same sign scenario.

---

### Official Review · Reviewer_Pmty · 2022-10-25

**Confidence:** 4
**Correctness:** 2
**Technical Novelty And Significance:** 2
**Empirical Novelty And Significance:** 2
**Recommendation:** 5

**Clarity, Quality, Novelty And Reproducibility:**

This paper proposes a novel defense method for backdoor attacks in FL, however, the experimental verification of this method is not enough. Thus, we can not determine whether this is a general method or only suitable for some special cases.

**Strength And Weaknesses:**

Strength:
1. This paper proposed a simple yet effective approach to mitigate backdoor attacks in FL. compared with previous robust aggregation methods, the proposed method can defend against a much stronger attack, i.e., an edge-case attack.
2. This paper gives robustness analyses for both AND mask and trimmed mean.

Weakness:
1. Only one backdoor attack in FL (edge-case attack) is tested in the experiments, which cannot validate the proposed method is a general defense method for backdoor attacks in FL. What about other frequently adopted backdoor attacks in FL, such as [1], and the experimental setting adopted in [2].
2. According to the federated learning setup in the experiments, it adopts an iid client setting. In the case of non-iid clients, will the proposed method of masking inconsistent gradients still work?
3. Masking inconsistent gradients will also sacrifice a lot of benign accuracies. However, according to the experiment results, the benign accuracy is hardly affected, could you give a more clearer explanation for this? Besides, the benign accuracy is also very low for FedAvg, which is very strange since usually, the benign accuracy for cifar10 in FL is at least over 80%.
[1] Bagdasaryan E, Veit A, Hua Y, et al. How to backdoor federated learning[C]//International Conference on Artificial Intelligence and Statistics. PMLR, 2020: 2938-2948.
[2] Panda A, Mahloujifar S, Bhagoji A N, et al. SparseFed: Mitigating Model Poisoning Attacks in Federated Learning with Sparsification[C]//International Conference on Artificial Intelligence and Statistics. PMLR, 2022: 7587-7624.

**Summary Of The Paper:**

This paper proposed a method to mitigate backdoor attacks in federated learning by focusing on invariant directions of gradients and avoiding selecting directions that favor malicious clients.

**Summary Of The Review:**

This paper is marginally below the acceptance threshold since although the authors proposed a simple yet effective approach to mitigate backdoor attacks in FL, we can not determine whether this is a general method or only suitable for some special cases.

---

> ### Author Response · Authors · 2022-11-17
> **Response to Reviewer Pmty**
>
> Thanks for the comments. We have clarified our data setting and accuracies in the response. We have also added additional empirical evaluations with more attacks and defenses to the revised version.
>
> &nbsp;
>
> **Only one backdoor attack in FL (edge-case attack) is tested in the experiments, which cannot validate the proposed method is a general defense method for backdoor attacks in FL. What about other frequently adopted backdoor attacks in FL, such as [1], and the experimental setting adopted in [2].**
>
> We believe our experiments are quite extensive. Our experiments include two backdoor attacks because model replacement and train-and-scale strategies from [1] are also considered in the edge-case backdoor framework (Section 2 of [2]). For the train-and-scale strategy, we scaled malicious updates to the norm clipping threshold. We don’t assume additional anomaly detectors, so the train-and-scale strategy, which is particularly designed to circumvent anomaly detectors, is not considered.
>
> In addition, we evaluated our approach with three types of triggers: the semantic trigger on CIFAR-10, word trigger on Twitter, and value trigger on Phishing.
>
> In the post-rebuttal version (Appendix D.4), we have completed additional evaluations of our defense against distributed backdoor attacks [3], adaptive backdoor attacks that are particularly designed against our defense, colluding attacks from SparseFed [4], and adaptive colluding attacks. Our results show that our approach remains effective and loses no more than 5% robustness against the four improved attacks. For example, our approach decreases the average accuracy on CIFAR-10 backdoor samples from 0.717 to 0.001 against edge-case backdoor attacks and decreases the average accuracy on CIFAR-10 backdoor samples from 0.717 to 0.049 against improved backdoor attacks.
>
> &nbsp;
>
> **According to the federated learning setup in the experiments, it adopts an iid client setting. In the case of non-iid clients, will the proposed method of masking inconsistent gradients still work?**
>
> All three datasets follow non-iid data partitions, although feature spaces and label spaces are the same across clients. CIFAR-10 and Twitter datasets follow standardized data partitions (open-source [5, 6]). The phishing dataset contains emails from different institutions and is non-iid.
>
> &nbsp;
>
> **Masking inconsistent gradients will also sacrifice a lot of benign accuracies. However, according to the experiment results, the benign accuracy is hardly affected, could you give a more clearer explanation for this?**
>
> Inconsistent gradients are caused by non-invariant features. Typical examples of inconsistent features include color features in images [7]. Avoiding inconsistent features does not significantly reduce accuracies because models can predict well using only invariant features (e.g., shape features) [7, 8]. Importantly, invariant features have consistent benefits across many clients and are, therefore, not masked out.
>
> &nbsp;
>
> **Besides, the benign accuracy is also very low for FedAvg, which is very strange since usually, the benign accuracy for cifar10 in FL is at least over 80%.**
>
> Our implementation is based on the Flute library, and the benign accuracy on CIFAR-10 is close to the reported result in [9]. Different libraries may have different data configurations, which lead to different accuracies.
>
> &nbsp;
>
> **Reference**
>
> [1] Bagdasaryan, Eugene, et al. "How to backdoor federated learning." International Conference on Artificial Intelligence and Statistics. PMLR, 2020.
>
> [2] Wang, Hongyi, et al. "Attack of the tails: Yes, you really can backdoor federated learning." Advances in Neural Information Processing Systems 33 (2020): 16070-16084.
>
> [3] Xie, Chulin, et al. "Dba: Distributed backdoor attacks against federated learning." International Conference on Learning Representations. 2019.
>
> [4] Panda, Ashwinee, et al. "SparseFed: Mitigating Model Poisoning Attacks in Federated Learning with Sparsification." International Conference on Artificial Intelligence and Statistics. PMLR, 2022.
>
> [5] He, Chaoyang, et al. "Fedml: A research library and benchmark for federated machine learning." arXiv preprint arXiv:2007.13518 (2020).
>
> [6] Caldas, Sebastian, et al. "Leaf: A benchmark for federated settings." arXiv preprint arXiv:1812.01097 (2018).
>
> [7] Arjovsky, Martin, et al. "Invariant risk minimization." arXiv preprint arXiv:1907.02893 (2019).
>
> [8] Sun, Mingjie, et al. "Can Shape Structure Features Improve Model Robustness under Diverse Adversarial Settings?." Proceedings of the IEEE/CVF International Conference on Computer Vision. 2021.
>
> [9] Dimitriadis, Dimitrios, et al. "Flute: A scalable, extensible framework for high-performance federated learning simulations." arXiv preprint arXiv:2203.13789 (2022).

---

> > ### Comment · Reviewer_Pmty · 2022-11-24
> > **Acknowledge the rebuttal**
> >
> > I would like to thank the authors for clarifying the concerns I raised. I would like to keep my score, as my major concern still remains: the framework should be evaluated against different types of existing backdoors, besides edge-case backdoor attacks.

---

> > > ### Author Response · Authors · 2022-11-26
> > > **Response to Reviewer Pmty**
> > >
> > > Thanks for replying. We would like to remind the reviewer that, as mentioned in the previous response, the post-rebuttal version includes 4 more attacks (Appendix D.4) in addition to the attack strategies in [1, 2]. The two works [1, 4] suggested by the reviewer are included. Specifically, the 4 backdoor attacks are:
> > >
> > > **1. Distributed backdoor attacks** [3]: Distributed backdoor attacks split triggers into pieces and let each malicious clients poison their data using one trigger piece. During testing, adversaries use full triggers without splitting to attack models.
> > >
> > > **2. Colluding backdoor attacks** [4]: Colluding adversaries let malicious clients report the same malicious update to servers.
> > >
> > > **3. Adaptive backdoor attacks**: Adaptive adversaries know our defense and zero the dimensions where signs of malicious updates conflict with the majority signs of benign updates. Then, adversaries scale up the remaining elements such that update magnitudes remain the same.
> > >
> > > **4. Adaptive colluding backdoor attacks**: Adversaries combine colluding backdoor attacks and adaptive backdoor attacks.
> > >
> > > Our results show that our approach remains effective and loses no more than 5% robustness against the four improved attacks. For example, our approach decreases the average accuracy on CIFAR-10 backdoor samples from 0.717 to 0.001 against edge-case backdoor attacks and decreases the average accuracy on CIFAR-10 backdoor samples from 0.717 to 0.049 against improved backdoor attacks.
> > >
> > > We would be happy to include more attacks as the reviewer suggests.
> > >
> > > &nbsp;
> > >
> > > **Reference**
> > >
> > > [1] Bagdasaryan, Eugene, et al. "How to backdoor federated learning." International Conference on Artificial Intelligence and Statistics. PMLR, 2020.
> > >
> > > [2] Wang, Hongyi, et al. "Attack of the tails: Yes, you really can backdoor federated learning." Advances in Neural Information Processing Systems 33 (2020): 16070-16084.
> > >
> > > [3] Xie, Chulin, et al. "Dba: Distributed backdoor attacks against federated learning." International Conference on Learning Representations. 2019.
> > >
> > > [4] Panda, Ashwinee, et al. "SparseFed: Mitigating Model Poisoning Attacks in Federated Learning with Sparsification." International Conference on Artificial Intelligence and Statistics. PMLR, 2022.

---

### Official Review · Reviewer_uH18 · 2022-10-27

**Confidence:** 4
**Clarity, Quality, Novelty And Reproducibility:** See the above strength and weakness.
**Correctness:** 3
**Technical Novelty And Significance:** 1
**Empirical Novelty And Significance:** 1
**Recommendation:** 3

**Strength And Weaknesses:**

Strengths

- FL is vulnerable to backdoor attacks

- A new aggregation rule is proposed.

Weaknesses

- Many robust aggregation rules have been proposed. This paper's proposed aggregation is just some variant of existing one, e.g., trimmed-mean. Therefore, the paper's novelty is limited.

- Seems like only edge-case backdoor attacks are evaluated. This represents a quite limited backdoor case. Other backdoor attacks should be evaluated.

- Some simple baselines are evaluated. Strong and more recent defenses (appeared in AI, security, and system venues) should be evaluated.

- Adaptive attacks are not considered.

**Summary Of The Paper:**

The paper proposes a new method to defend against backdoor attacks. The proposed method is basically a new aggregation rule. Edge-case backdoor attacks are considered, and multiple simple baselines are evaluated.

**Summary Of The Review:**

The paper has limited novelty, and experiments are also limited.

---

> ### Author Response · Authors · 2022-11-17
> **Response to Reviewer uH18**
>
> Thanks for the comments. We have clarified the novelty of our work and the attack strategies in the response. We have also added additional empirical evaluations with more attacks and defenses to the post-rebuttal version.
>
> &nbsp;
>
> **Many robust aggregation rules have been proposed. This paper's proposed aggregation is just some variant of existing one, e.g., trimmed-mean. Therefore, the paper's novelty is limited.**
>
> We respectfully disagree! The novelty of our work is two-fold. First, we show that focusing on invariant directions in the model optimization trajectory (using the AND-mask) can help defend against backdoor attacks. Neither the perspective of invariance directions nor the specific implementation using the AND-mask has not yet been explored in previous defenses against federated backdoor attacks.
>
> Second, we analyze the failure mode of the AND-mask and enhance it with the trimmed-mean estimator. We further demonstrate the complementary relationship between the AND-mask and the trimmed-mean estimator. The failure modes and this complementary relationship have not yet been discovered in prior works.
>
> Importantly, Table 1 in Section 6.2 further shows that a straightforward combination of existing approaches (i.e., Krum + trimmed-mean) does not succeed.
>
> &nbsp;
>
> **Seems like only edge-case backdoor attacks are evaluated. This represents a quite limited backdoor case. Other backdoor attacks should be evaluated. Adaptive attacks are not considered.**
>
> We believe our experiments are quite extensive. Our experiments include two backdoor attacks because model replacement and train-and-scale strategies from [1] are also considered in the edge-case backdoor framework (Section 2 of [2]). For the train-and-scale strategy, we scaled malicious updates to the norm clipping threshold. We don’t assume additional anomaly detectors, so the train-and-scale strategy, which is particularly designed to circumvent anomaly detectors, is not considered.
>
> In addition, we evaluated our approach with three types of triggers: the semantic trigger on CIFAR-10, word trigger on Twitter, and value trigger on Phishing.
>
> In the post-rebuttal version (Appendix D.4), we have completed additional evaluations of our defense against **4 more backdoor attacks**: distributed backdoor attacks [3], adaptive backdoor attacks that are particularly designed against our defense, colluding backdoor attacks [4], and adaptive colluding backdoor attacks. Our results show that our approach remains effective and loses no more than 5% robustness against the four improved attacks. For example, our approach decreases the average accuracy on CIFAR-10 backdoor samples from 0.717 to 0.001 against edge-case backdoor attacks and decreases the average accuracy on CIFAR-10 backdoor samples from 0.717 to 0.049 against improved backdoor attacks.
>
> &nbsp;
>
> **Some simple baselines are evaluated. Strong and more recent defenses (appeared in AI, security, and system venues) should be evaluated.**
>
> We have added additional evaluations with FoolsGold [5], RFA [6], and SparseFed [4] to Section 6.2 of the post-rebuttal version. Our approach is more robust than the recent defenses by 6% - 99%.
>
> &nbsp;
>
> **Reference**
>
> [1] Bagdasaryan, Eugene, et al. "How to backdoor federated learning." International Conference on Artificial Intelligence and Statistics. PMLR, 2020.
>
> [2] Wang, Hongyi, et al. "Attack of the tails: Yes, you really can backdoor federated learning." Advances in Neural Information Processing Systems 33 (2020): 16070-16084.
>
> [3] Xie, Chulin, et al. "Dba: Distributed backdoor attacks against federated learning." International Conference on Learning Representations. 2019.
>
> [4] Panda, Ashwinee, et al. "SparseFed: Mitigating Model Poisoning Attacks in Federated Learning with Sparsification." International Conference on Artificial Intelligence and Statistics. PMLR, 2022.
>
> [5] Fung, Clement, Chris JM Yoon, and Ivan Beschastnikh. "The limitations of federated learning in sybil settings." 23rd International Symposium on Research in Attacks, Intrusions and Defenses (RAID 2020). 2020.
>
> [6] Pillutla, Krishna, Sham M. Kakade, and Zaid Harchaoui. "Robust aggregation for federated learning." IEEE Transactions on Signal Processing 70 (2022): 1142-1154.

---

### Author Response · Authors · 2022-11-17
**General Response**

We thank all the reviewers for their helpful comments and suggestions. In the response, we have clarified the novelty of our approach and the broad applicability of our robustness guarantees. We would also like to highlight the simplicity of our defenses, which are yet shown to be highly effective.

As the reviewers suggested, we have included additional experiments in the post-rebuttal version. Specifically, we have considered **four more attack strategies**: adaptive attacks that are particularly designed against our defense, colluding attacks [1], adaptive colluding attacks, and distributed backdoor attacks [2] (Appendix D.4). Our results show that our approach remains effective and loses no more than 5% robustness against the four improved attacks. For example, our approach decreases the average accuracy on CIFAR-10 backdoor samples from 0.717 to 0.001 against edge-case backdoor attacks and decreases the average accuracy on CIFAR-10 backdoor samples from 0.717 to 0.049 against improved backdoor attacks.

In addition, we have compared our approach with **three state-of-the-art defenses**: FoolsGold [3], RFA [4], and SparseFed [1] (Section 6.2). Our approach is more robust than the recent defenses by 6% - 99%.

The two failure modes in Figure 1 are also empirically verified (Appendix D.5).

&nbsp;

**Reference**

[1] Panda, Ashwinee, et al. "SparseFed: Mitigating Model Poisoning Attacks in Federated Learning with Sparsification." International Conference on Artificial Intelligence and Statistics. PMLR, 2022.

[2] Xie, Chulin, et al. "Dba: Distributed backdoor attacks against federated learning." International Conference on Learning Representations. 2019.

[3] Fung, Clement, Chris JM Yoon, and Ivan Beschastnikh. "The limitations of federated learning in sybil settings." 23rd International Symposium on Research in Attacks, Intrusions and Defenses (RAID 2020). 2020.

[4] Pillutla, Krishna, Sham M. Kakade, and Zaid Harchaoui. "Robust aggregation for federated learning." IEEE Transactions on Signal Processing 70 (2022): 1142-1154.

---

### Decision · Program_Chairs · 2023-01-20

**Decision:**

Reject

**Justification For Why Not Higher Score:**

The paper has weak baselines (some of the included baselines seem to perform poorly compared to standard benchmarks) and the novelty of the method is not clear to reviewers.

**Justification For Why Not Lower Score:**

N/A

**Metareview: Summary, Strengths And Weaknesses:**

This work proposes a defense against federated learning backdoor attacks. The key idea is to force the model optimization trajectory to focus on the gradient directions that are invariant to gradient directions that favor malicious clients. Particularly, the authors consider the consistency of the client model update as an estimation of the invariance, where the AND-mask and Trimmed mean are used together to measure the sign consistency.

All reviewers felt that the novelty is limited, that in the proposed method the difference between mean and trimmed-mean is not clear. There are questions about the baseline implementations.

The authors should include more baselines (answer the questions about the performance of existing baselines) and explain clearly what is novel about their method, what is non-trivial about their proposed solution.


```
The rebuttal did not address the main concerns. Discussion with one of the reviewers during the rebuttal period:

Dear AC,

The rebuttal does not address my concerns.

According to the new results in Appendix D.5, the gradient signs are not that different between benign and malicious updates (<= 0.02 for Twitter and Phishing datasets). Also, the difference between Mean and Trimmed-mean is not obvious. The paper says "trimmed-means’ advantages vanish in inconsistent dimensions". However, according to Table 5, on FEMNIST and Phishing, Trimmed-mean still has slightly lower number of different gradient signs. The results do not seem to validate the motivation in Figure 1. It is unclear why the method works.

For baselines FoolsGold and RFA, according to my knowledge, they can also reduce the ASR to less than 10% in the single-shot attack. The results reported in the paper are higher than expected. Another baseline FLTrust, which is not compared in the paper (mentioned in the rebuttal), has similar or better performance than the above baselines.

The adaptive attack evaluated in the paper is not strong. It only tries to zero the gradient dimensions instead of proactively matching the gradient signs between malicious and benign updates. It is still unclear how robust the proposed method is against strong adaptive attacks.

The argument in the rebuttal against the novelty is not convincing. The proposed method is also a straightforward combination of two existing techniques as shown in Algorithm 1. The explanation on why the proposed method works is unclear as the empirical results do not seem to validate the paper's motivation.```